# Recent Advances in Asymmetric Iron Catalysis

**DOI:** 10.3390/molecules25173889

**Published:** 2020-08-26

**Authors:** Alessandra Casnati, Matteo Lanzi, Gianpiero Cera

**Affiliations:** 1Laboratoire des Systèmes Complexes en Synthèse et Catalyse, Institut de Science et d’Ingénierie Supramoléculaires, Université de Strasbourg &CNRS, 8 Allèe Gaspard Monge, BP 70028, F-67083 Strasbourg, France; casnati@unistra.fr; 2Laboratoire de Chemie Moléculaire (UMR CNRS 7509), Université de Strasbourg, ECPM 25 Rue Becquerel, 67087 Strasbourg, France; lanzi@unistra.fr; 3Dipartimento di Scienze Chimiche, della Vita e della Sostenibilità Ambientale, Università di Parma, Parco Area delle Scienze 17/A, I-43124 Parma, Italy

**Keywords:** iron, enantioselective, catalysis

## Abstract

Asymmetric transition-metal catalysis represents a fascinating challenge in the field of organic chemistry research. Since seminal advances in the late 60s, which were finally recognized by the Nobel Prize to Noyori, Sharpless and Knowles in 2001, the scientific community explored several approaches to emulate nature in producing chiral organic molecules. In a scenario that has been for a long time dominated by the use of late-transition metals (TM) catalysts, the use of 3d-TMs and particularly iron has found, recently, a widespread application. Indeed, the low toxicity and the earth-abundancy of iron, along with its chemical versatility, allowed for the development of unprecedented and more sustainable catalytic transformations. While several competent reviews tried to provide a complete picture of the astounding advances achieved in this area, within this review we aimed to survey the latest achievements and new concepts brought in the field of enantioselective iron-catalyzed transformations.

## 1. Introduction

Transition metal catalysis represents a powerful tool for the construction of highly functionalized molecules in organic synthesis. Among others, iron-catalyzed reactions [1,2,3,4,5] have recently become the most attractive alternative to the well-established palladium [6], rhodium [7], iridium [8] and ruthenium ones [9]. As a result of its low cost, compared to other d-metals, iron represents the most prominent metal for innovative and sustainable catalysis [10]. In addition, life evolution has integrated iron compounds in biological systems, thus resulting in low toxicity and therefore allowing high concentrations (>1300 ppm) of this metal in pharmaceuticals [11,12]. The chemistry of iron complexes is vast. Its position in the periodic table, just above ruthenium, justifies its wide applicability, mostly due to the various accessible oxidation states (from −II to +VI). Low-valent iron complexes proved to be active for catalytic nucleophilic substitutions, hydrogenation/hydrosilylation and cycloisomerization reactions [13,14]. Parallelly, the more common +II and +III iron species exhibit Lewis acidity that has been exploited in the electrophilic aromatic substitutions, such as Friedel–Crafts reactions [15]. Radical reactions and two electron couplings are also feasible under iron catalysis. The latter, in particular, involves a Fe(I)/Fe(III) manifold that parallels [16], in terms of reactivity, those operating in palladium-catalyzed cross-couplings [17].

The stereochemical control of chiral centers during chemical reactions is fundamental to access enantiopure compounds and thus avoiding tedious purifications. As a matter of fact, asymmetric catalysis with transition metals proved to be one of the most economical, environmentally friendly and efficient approaches for the synthesis of enantiomerically pure molecules [18].

Iron-catalyzed enantioselective reactions historically were inspired by biological processes. Iron is present in several enzymes, such as for instance, the heme containing cytochrome P450, known for its capability to promote C‒H oxygenations [19] and carbene transfer reactions [20]. The chemist’s need to emulate nature led to the development of a remarkable number of iron-catalyzed enantioselective methodologies, initially using chiral iron porphyrin complexes [21,22,23]. Major advances were subsequently achieved with the introduction of chiral ligands such as oxazolines (i), NHCs (ii) or phosphine-based ligands (iii) that for a long time represented the most commonly employed approach for highly enantioselective transformations. More recently, new strategies have been introduced by the chemical community. For example, the use of achiral ligands, asymmetrically coordinated around the metal center, opened access to catalytically active chiral-at-metal iron complexes (iv). Additionally, asymmetric induction could be also induced by a second coordination sphere (v), usually constituted by bio macromolecules, namely enzymes or DNAs.

The aim of this review is to provide an overview of the last advances in enantioselective iron-catalyzed reactions describing their different working-modes (i–v) to induce enantioselectivity. The last three years (2018, 2019 and 2020) will be covered with few references from the previous two years.

## 2. Chiral Oxazolines

Chiral oxazolines are a family of ligands that has been extensively employed to develop enantioselective transformations. In the past three decades, they received remarkable attention due to their efficiency and versatility [24,25,26]. Examples including early transition metals, such as iron, are however more recent [27]. A chiral spiro bisoxazoline iron complex was found catalytically active in promoting the intramolecular enantioselective cyclopropanation of indoles [28]. This methodology gives access to polycyclic compounds **2** with two contiguous all-carbon quaternary stereogenic centers. Chiral ligand **L1** was also active in the presence of copper precatalysts. Different α-aryl-α-diazoesters were evaluated in the intramolecular cyclopropanation reaction providing high yields and enantioselectivities (Scheme 1).

The cyclopropane-containing final product **2**, due to its donor-acceptor feature, allowed multiple additional transformations to access complex polycyclic scaffolds. The same class of chiral ligands was selected for iron-catalyzed asymmetric Si‒H insertions of α-diazoesters [29]. In particular, chiral hexamethyl-1,1′-spirobiindane-based bisoxazoline ligands (**L2**, HMSI-BOX) proved to be superior to other bisoxazoline ligands in ensuring high levels of stereoinduction (Scheme 2).

Here, the product selectivity is determined by a concerted Si‒H bond oxidative addition/hydride insertion sequence. The relative Gibbs free-energy barriers associated to the two transition states at alternative spin states were measured via DFT calculations. The quintet-state spin configuration resulted as the lowest in terms of energy barrier, and the stereocontrol is induced by steric repulsions present in one of the two transition states between the phenyl group on the ligand and ester moieties of starting materials **3**. The substrate scope was investigated and a variety of alkyl α-diazoarylacetates **3** was successfully silylated with Et_3_SiH or Me_2_PhSiH with high enantioselectivities (up to 90% *ee*). Another strategy to obtain chiral organosilanes, involved alkene hydrosilylation. In 2018 Lu’s research group reported a Markovnikov-selective and enantioselective hydrosilylation of challenging unactivated terminal alkenes **6** [30]. Chiral oxazolineiminopyridine iron complexes (**L3**·FeCl_2_) were selected as promising catalysts to promote the desired reactivity. The presence of *ortho* substituents on the imine fragment of the ligand was crucial to obtain high branched vs. linear selectivity. Particularly, the presence of large CHPh_2_ groups at both the *ortho* positions of the aryl ring was essential to obtain the transformation with excellent enantioselectivities. Subsequently, the authors explored the substrate scope; remarkably in all the examples they obtained high regioselectivities (>96/4 branched vs. linear) and enantiocontrol (>97% *ee*). Different functional groups were well tolerated, and the late stage functionalization of bioactive molecules, such as naproxen and ibuprofen, was accomplished (**8f,g**; Scheme 3).

The proposed mechanism goes through the formation of an iron silyl intermediate (**A**) rather than an iron-hydride species due to the easy β-hydride elimination that eventually occurs with in-situ formed secondary alkyl-iron compounds. The alkene inserts into the iron-silyl bond (**C**) and then reacts with PhSiH_3_ to regenerate the Fe‒Si active species **A**, affording the final product **8** (Scheme 4).

## 3. *N*-Heterocyclic Carbenes (NHCs)

*N*-Heterocyclic carbenes (NHCs) are currently in use as both nucleophilic organocatalysts [31] and as electron-rich neutral ligands in combination with several transition metals [32]. The large success of achiral NHCs ligands led to the development of chiral equivalents as stereodirecting ligands for enantioselective transformations [33]. The construction of their scaffold is easily achieved from imidazolium salt precursors, thus favoring the access to a vast library of tailor-made chiral compounds. In 2017 Ackermann’s group reported an iron-NHC catalyzed enantioselective C‒H alkylation of indoles **9** [34]. Among all the NHCs ligands tested, the *meta*-substituted *N*,*N*′-diarylated NHC proved to be the most active, providing high levels of enantioselectivities. A *meta*-1-adamantyl derivative **L4** ensured an *er* of 96:4 for the C‒H alkylation of indole **9** with alkenyl ferrocenes, significant building blocks in material sciences and bioinorganic chemistry [35]. The optimized reaction conditions allowed us to functionalize diversely substituted indoles with a variety of styrenes **10** including styryl ferrocenes and ruthenocenes, among others (Scheme 5).

To elucidate the mechanism of this highly positional selective C‒H alkylation, first isotopically labeled substrates were investigated. Deuterium was transferred to the terminal position of the alkene supporting an inner sphere C‒H activation. This mode of action can be rationalized with the C‒H cleavage occurring by ligand-to-ligand H-transfer (**G**, LLHT) or by C‒H oxidative addition. The proposed catalytic cycle starts with the reduction of the iron(III) precatalyst via β-hydrogen elimination, triggered by the action of CyMgCl. The so-formed iron-NHC coordinates in a reversible manner the alkene (**D**, zero^th^-order dependence). The substrate undergoes C‒H metalation via LLHT (**G**) [36]. Then the stereo-determining migratory insertion takes place and it is induced by ligation of the substrate **9**. Here, the stereoinduction is controlled by dispersive secondary interactions [37]; one side of the double bond suffers from steric clashes with the *meta*-bulky group of the NHC ligand, thus favoring a unilateral coordination to the metal center (Scheme 6).

## 4. P-Based-Ligands

Phosphine chiral ligands covered an important role in enantioselective transition metal-catalyzed reactions. Despite a widespread application of iron catalysis in cross-couplings [38], an enantioselective protocol remained, for a long time, elusive [39]. In this context, recently Nakamura and co-workers pioneered this field reporting the first example of enantioselective iron-catalyzed Kumada coupling [40]. Based on these studies, last year the same group described an enantioconvergent construction of C(sp^3^)‒C(sp^3^) bonds through an iron-catalyzed Suzuki-Miyaura reaction [41]. Chiral diphosphines were found as suitable ligands for the conversion of *tert*-butyl α-bromo propionate **12** in the presence of lithium aryl boronates **13**. Bidentate aliphatic and benzene bridged chiral ligands did not provide any stereocontrol, albeit ensuring high reaction yields. Contrary, excellent yield and high enantioselectivity were achieved with *C*1-symmetric quinoxaline **L5**. The addition of TFA at the end of the reaction enabled the isolation of the optical active α-arylated carboxylic acid through a one-pot procedure (Scheme 7).

Mechanistic investigations were performed to elucidate the enantioselective induction showed by (*R*,*R*)-quinoxP* **L5**. Complex FeCl_2_/(*R*,*R*)-quinoxP* **I**, synthetized separately, was reacted with phenyl boronate **13a** (5 equiv) and MgBr_2_ (1 equiv) to form complex **J**, within 2 h. This manifold was compared to the one constituted by the FeCl_2_/(*R*,*R*)-BenzP* complex, previously employed for the enantioselective Kumada reaction. Indeed, after 62 hrs, only 60% of the starting iron complex was consumed. This finding highlighted how the electron withdrawing nature of the ligand renders the iron center more electrophilic, hence accelerating the transmetalation step. On the basis of several experimental evidences, a plausible catalytic cycle was proposed. The in-situ formed iron(I) complex **H** can reduce the alkyl bromide to form an alkyl radical and the iron(II) species **I**. The subsequent transmetalation provides intermediate **J**, which through radical recombination forms an iron(III) intermediate **K**. DFT calculations proved this step as enantio-determining in the catalytic cycle. Finally, the desired product **14a** is formed after reductive elimination from complex **K** and eventual acidic hydrolysis (Scheme 8).

Directing group (DG)-assisted C‒H activation methodologies represent an important synthetic tool with the aim to functionalize organic molecules in a site-selective manner [41]. In this context, triazole TAM DGs introduced by the group of Ackermann are gaining increasing attention in promoting iron-catalyzed C‒H functionalizations [4]. Inspired by earlier contributions [42,43], Schmiel and Buthenshön disclosed an enantioselective iron-catalyzed C‒H arylation [44], setting the stage for the synthesis of planar chiral ferrocenamides **17**, **18** (Scheme 9).

The use of commercially available *P*,*P*-chiral ligand **L6** allowed for the synthesis of **17**, **18** in high yields. Although the enantioselection is just moderate, this represents the first example of asymmetric iron-catalyzed C‒H functionalization using bidentate DGs.

Chiral phosphoric acids are largely used in enantioselective transformations either in combination with a metal or under metal-free conditions [45]. Their synthesis, which suffers from tedious multistep synthetic sequences, principally requires the use of enantiomerically pure *C*1-chiral BINOL structures. A step-economical approach for BINOLs, starting from commercially available 2-naphthols **19**, was achieved by Katsuki, Pappo and Bryliakov under iron catalysis [46,47,48]. Recently, Pappo and co-workers developed an iron-catalyzed chiral-anion strategy for the enantio-construction of atropoisomeric C(sp^2^)‒C(sp^2^) bonds (Scheme 10) [49].

Various 3,3′-disubstituted phosphoric acids were found active and promoted the enantioselective transformation, albeit with low *ee*s, suggesting the formation of a labile chiral iron intermediate. Indeed, using catalyst (**L7**)_3_Fe, previously prepared from the corresponding chiral phosphoric acid and Fe(ClO_4_)_3_ in the presence of di-*tert*-butyl peroxide, the desired BINOL was delivered in 86% yield and 94:6 *er*. Since the metal center in complex **L7** does not present any vacant coordination site, the reaction was conducted with different ratios of iron and ligands (1:1; 1:2 and 1:3) in order to identify the catalytic active species during the transformation. The low enantioselectivity demonstrated with 5 mol% of ligand with respect to 10 or 15 mol%, suggested the presence of a catalytically operative (**L7**)_2_Fe complex, which undergoes ligand exchange with 2-naphthol **19** to form an (**Lx**)_3_Fe species that finally enabled the transformation. A broad scope for chiral 7,7′-BINOLs **20** was reported with high yields and good to excellent enantioselectivities. These latter were found affected by the electronic nature of substrates; thus, electron-poor reagents were converted with higher enantioselectivity compared to electron-rich ones.

The erosion of the enantiopurity of the products, in the presence of catalytic amounts of FeCl_3_ was hypothesized to be promoted by a reversible single electron transfer (SET) process involving the binaphthyl metal complex that generated a delocalized biaryloxy radical. These evidences supported the hypothesis of a radical anion coupling mechanism to be operative [50]. On the basis of these experimental results, the catalytic cycle was proposed. Coordination of peroxide to iron(III) forms **L** that subsequently undergoes ligand exchange with **19** and peroxide-mediated bond cleavage to provide intermediate **M**. Transfer of the electron density from the naphtholate ligand (**N**) enabled a radical-anion coupling with a second naphthol unit **19**, leading to **O**. Finally, radical abstraction operated by the peroxide (**P**), followed by transmetallation, releases product **20** restoring the iron catalyst (Scheme 11).

More recently, also chiral diphosphine oxide ligands were found as suitable ligands for the enantioselective synthesis of BINOLs. Hence, Ishihara and co-workers reported a stable diphosphino oxide iron(II) complex, which acts as a better oxidant for the radical-anion coupling [51]. Various chiral diphosphines ligands were tested in the presence of Fe(OTf)_2_ and *tert*-butyl peroxide. BINOLs-based backbone ligands failed to induce any stereocontrol, while high yields and enantioselectivities were observed with ligands **L8** and **L9**. The in-situ formed **L8**·Fe(OTf)_2_ complex was found active for the oxidative coupling of a large family of 3,7- and 6-substituted 2-naphthols **19** (Scheme 12).

To shed light on the mode of action of the iron catalysis, mechanistic experiments were conducted. Interestingly, an enantioselective non-liner positive effect was observed for the reaction with ligand **L8**. In addition, a stoichiometric reaction conducted in the presence of **L8**·Fe(OTf)_2_ with 1 equiv of *t*BuOOH, led to the desired product in moderate yield (47%), suggesting the consumption of part of the oxidant to generate the active species Fe(III) from Fe(II). This finding confirms a Fe(III)/Fe(IV) catalytic manifold as previously described by Pappo and Katsuki [49,50].

## 5. Chiral-to-Metal

Commonly, enantioselective transformations are designed using a catalytic system composed by a metal center and cost-effective chiral ligands. In the last years, appealing alternatives have been developed and involve the use of readily available achiral ligands arranged into an asymmetric octahedral coordination sphere around the metal. Despite various chiral-at-metal catalysts have been reported for the 4d and 5d transition metals [52,53], 3d chiral complexes remain less investigated. Meggers and co-workers pioneered this field presenting the synthesis and the catalytic activity of chiral-to-metal iron catalysts [54]. The synthesis of iron compounds was accomplished by an electrolysis reaction of the corresponding in situ-generated PyNHC silver salt, in MeCN, with an iron sacrificial anode. Thus, the racemic iron intermediate **L10**·Fe(PF_6_)_2_(MeCN)_2_ was delivered in good yield. Separation of diastereoisomers, obtained by complexation with a chiral oxazoline and subsequent treatment with NH_4_PF_6_, yielded d-(*S*)-**L10**·Fe(PF_6_)_2_(MeCN)_2_ (**Λ-Fe**) and l-(*R*)-**L10**·Fe(PF_6_)_2_(MeCN)_2_ (**Δ-Fe**) chiral catalysts. These bench stable Lewis acids resulted in being highly enantiopure (>99:1) and showed mirror-imagine CD spectra proving the presence of the two corresponding enantiomers (Scheme 13).

Chiral Lewis acid iron(II) complexes **Λ-Fe** and **Δ-Fe** demonstrated peculiar catalytic activities. The asymmetric conversion of phenylglyoxal **21** to mandelate ester **22** proceeded in high yield and enantioselectivity (Scheme 14a). As expected, **Δ-Fe** catalyzed the formation of the corresponding mirror-imagine of **22** with comparable yield and stereoselection. These complexes were also found active in promoting the asymmetric Nazarov reaction of **23** providing the formation of (1*R*, 2*S*)-**24** in high *dr* and *ee* (Scheme 14b). Although the precise mechanism is unknown, the high enantiocontrol displayed by the catalysts, suggested iron-substrate interactions to occur into the chiral pocket.

More recently, Che and co-workers described the catalytic application of iron(II) complexes with metal- and ligand-chirality [55]. The reaction of different *N*_4_-type chiral ligands, such as *bis*(quinolyl)diamine with the Fe(OTf)_2_(MeCN)_2_ precatalyst, allowed for the synthesis of a family of bench stable iron complexes, where the octahedral coordination provided a metal chirality feature to the catalyst. The catalytic activity was subsequently demonstrated in FC-type alkylation reactions of indoles **26** with α,β-unsaturated acyl imidazoles **25** (Scheme 15).

The iron complex **L11**·Fe(OTf)_2_, bearing quinoline fused pentane rings, proved to be the best catalyst, leading to **27** in high yield and enantioselectivity. Interestingly, using 2,3-di-substituted substrates, *N*-alkylated indoles were obtained with comparable yields and stereoselection (**27d,e**, Scheme 15). The Fe(II) chiral catalyst **L11**·Fe(OTf)_2_ was also found active for the alkylation of electron-rich anilines **28** and pyrroles **29**. High enantioselectivities were obtained for the positional selective *C*-2 and *C*-3 alkylation of pyrroles, with these latter achieved in the presence of 2,5-disubstitued substrates **31b,c** (Scheme 16).

To shed light on the mechanism and on the origin of the enantioselectivity, DFT calculations were conducted. The lowest energy catalytic pathway is depicted below (Scheme 17) and begins with the coordination of **25** to the chiral metal center (**Q**). A bidentate coordination of **25** to the catalyst preserves the *Si*-prochiral face and enables the interaction of the indole to the *Re*-face. In addition, π–π stacking interactions between the indole and the quinoline ligand moieties reduce the energy of the transition state. The subsequent proton transfer and the exothermic tautomerization finally generate product **27** (Scheme 17).

## 6. Second Coordination Sphere

In transition metal catalysis, the catalytic activity of the metal center is often controlled by the ligands, which constitute the first coordination sphere. In this context, enantioselectivity is achieved by directing the incoming reagent to one prochiral side, by blocking, often sterically, the other one.

Enzymatic catalysis features hybrid catalysts composed of a catalytically active transition metal complex integrated in a biomolecular scaffold, usually a protein [56] or DNA [57]. In this case, chiral discrimination is induced by the second coordination sphere [58,59], with the chiral environment provided by bio macromolecules. In this way, biomolecular scaffolds can direct the reactivity to one prochiral face of the bounded reactant or favor the structure of one enantiomer’s transition state [60].

Iron plays a crucial role in enzymatic catalysis because most of the studies are focused on the modification of enzymes containing iron-heme complexes [61,62,63].

One of the undisputed leaders in this field is the 2018 Nobel laureate Frances Arnold [64,65]. In 2017 Arnold’s research group reported an enantioselective intermolecular benzylic C–H amination [66]. The active catalyst consists of the directed evolution of an iron containing enzyme based on cytochrome P450 monooxygenase. This enzyme is known to nature because it promotes the direct insertion of oxygen into C–H bonds [67]. By just modifying the cysteine axial ligand with a serine one, it is possible to obtain a class of modified cytochromes called P411s. This modification of the axial ligand of the heme cofactor increases the Fe(III)/Fe(II) reduction potential. These enzymes are able to catalyze nitrene transfer reactions (Scheme 18) [68].

To probe the selected transformation, 4-ethylanisole **32a** was reacted with tosyl azide with whole *E. coli* cells overexpressing P411. Other modifications (Enzyme P411_CHA_) were necessary in order to maximize the selectivity toward the desired product **34a** over the reduction of the nitrenoid. Beneficial mutations were identified in the active site of the enzyme that mediates the substrate binding. The scope of the reaction was then investigated. Substitutions in *para*, *meta* and *ortho*- position of the aromatic ring were well tolerated (Scheme 19, **34a–c**). Although the presence of electron-withdrawing groups decreased the reactivity, halogens substituents could be still successfully converted into the corresponding products with excellent enantioselectivity (**34d**). The enzyme active site could accommodate also larger substrates such as 4-propylanisole (**34e**), while tetraline and indane (**34f,g**) were suitable candidates for the enantioselective C–H amination, as well.

The same class of serine-ligated P411 enzymes was investigated for the catalytic carbene transfer to internal alkynes to afford cyclopropene products **37** as a single enantiomer. A proper engineering of the active site of the enzyme was fundamental in order to avoid the steric clashes between the liner π-system and the planar heme cofactor (Scheme 20) [69].

The scope of the reaction was then evaluated and aromatic internal alkynes bearing different substituents on the aromatic ring were tested. Other modifications were needed in order to expand the substrate scope due to the hypothesized specificity acquired from the first enzyme modification towards unsubstituted aromatic ring or electron-rich alkynes. Indeed, replacing a serine with a glycine in the active site, allowed one to increase the space for substituted aromatic rings obtaining from moderate to good results with a broad variety of all-substituted, as well as disubstituted, rings. Internal aliphatic alkynes proved to be more challenging substrates. Nevertheless, proper modifications revealed to be fundamental to allow their reactivity. Moreover, it was possible to tune the chemoselectivity to obtain either the C‒H insertion or the cyclopropenation. Ultimately, they found that this enzymatic platform allowed one to readily scale-up the reaction.

Very recently, the same research group proposed a methodology to access aliphatic primary amines via C(sp^3^)–H aminations [70]. The directed evolution of cytochrome P411 enzymes allowed to selectively functionalize benzylic and allylic C–H bonds providing a broad range of primary amines (**39** and **41**) through the formation of high-valent unprotected iron-nitrenoid intermediates [71]. Several mutations were introduced in the active site, granting the formation of the desired products. Secondary benzylic C–H bonds ensured the best yields with high enantioselectivities (Scheme 21a). This class of enzymes also allowed for the C–H amination of allylic bonds by an appropriate engineering of these former; the modifications were aimed at minimizing the possible side reactions involving the olefin. The optimized new enzyme provided high yields and selectivity for a different range of allylic substrates **41** (Scheme 21b).

Non-heme iron enzymes were recently found as feasible catalysts for nitrene transfer reactions. Hence, in 2019, Arnold’s research group described olefin aziridinations and nitrene C–H insertions [72]. α-Ketoglutarate-dependent iron enzymes were selected as proper tool to investigate the aziridination of styrene **42**. This family of enzymes binds iron through two histidines and one aspartate or glutamate. Of the three available coordination sites, two are occupied by the α-ketoglutarate and the third is required for the specific reactivity of the enzymes (often an iron-oxo intermediate). Due to the non-heme structure of the enzyme, the first coordination sphere of the metal can be modified with small molecules in order to tune the activity and the selectivity of the enzyme (Scheme 22).

Several small molecules related to α-ketoglutarate were investigated and it was found that the presence of a carboxylate was beneficial for the reactivity. The selected ligand species was acetate for its ubiquity and inexpensiveness. Of course, some mutations in the binding pocket were required to obtain high enantioselectivity for **44–46** (88% and 64% *ee* respectively, Scheme 22).

As mentioned above, the second coordination sphere interactions can involve DNA structures as a biomolecular scaffold around the metal complex, providing a chiral environment to develop enantioselective transformations. In 2016, Roefels and co-workers disclosed a DNA/cationic-iron-porphyrin-based catalytic system for the styrenes cyclopropanation via carbene transfer [73]. *Meso*-tetrakis(*N*-alkylpyridyl)porphyrins **L13**·FeCl, well-established DNA binders, were chosen as a model catalytic system to study the reactivity. Their binding properties depend on the type and length of the alkylic chain. Among all the porphyrins tested, those bringing 2-*N*-methyl pyridines gave the best enantioselectivity together with a marked acceleration effect in the presence of the DNA. The porphyrin’s interaction with DNA mostly occurs through groove binding in AT-rich areas. The groove binding makes the catalytic site more accessible to the substrate, being at the periphery of the structure of the DNA. Moreover, the iron porphyrins binding with the DNA forms hydrophobic pockets where the reagents are concentrated. For this reason, the concentration plays an important role to exploit the desired reactivity and the rate acceleration due to the DNA (Scheme 23).

## 7. Others

Despite notable advances obtained for the enantioselective synthesis of BINOL derivatives, strategies for chiral 2-amino-2′-hydroxy-1,1′-dinaphtyl (NOBIN) ligands are limited to multistep synthetic approaches [74]. Moreover, these protocols do not allow the presence of substitutions at 3,3′-positions, often important for the design of effective chiral catalysts [75]. To overcome these limitations, an enantioselective protocol for the synthesis of NOBIN derivatives **49** has been recently devised by Pappo and co-workers through a peculiar point-to-axial chiral transfer approach [76] (Scheme 24).

An iron(III)-catalyzed reaction between 2-naphthols **19** and chiral labeled 2-aminonaphthalenes **48** enabled the access to a separable mixture of diastereoisomers **49**. The reaction required the presence of trifluoroacetic acid without the need of any additional expensive chiral ligand. The unique possible side products of the reaction, BINOL and the **48** homocoupling product, were absent. This highlighted the chemoselectivity of the method, further suggesting the presence of a naphthoxyl radical intermediate. A large family of NOBIN derivatives **50** was efficiently accessible with moderate to high yields. The presence of substituents at the *C*-3 position of starting materials ensured higher enantioselectivities and allowed for the formation of a single product; this could be rationalized by the high steric hindrance present in the inner sphere transition state. The involvement of a naphthoxyl radical was confirmed by the complete racemization of *R*_a_-NOBIN with a catalytic amount of FeCl_3_ (10 mol%) within 5 h. The following removal of the chiral auxiliary afforded (+)-*R*_a_-**50e** and (−)-(*S*_a_)-**50e** in high enantioselectivity (up to 99% *ee*). The utility of the methodology was also reported with the large-scale synthesis of a NOBIN-backboned phosphoric acid **51** (Scheme 25).

Enantiopure chiral substrates were also used by Baik, Cook and co-workers to develop iron(III)-catalyzed stereospecific nucleophilic substitutions [77]. Preliminary studies on intermolecular amination reactions of secondary and tertiary alcohols demonstrated the efficiency of iron catalysts, which afforded the corresponding desired products in high yields. In contrast, an enantioselective stereoinverting pathway was observed for the intramolecular amination reaction. The Fe(SbF_6_)_3_ complex, formed in situ from FeCl_3_ and AgSbF_6_, catalyzed the stereoinverting amination reaction of enantiopure secondary alcohols **52** with high enantiospecificity and quantitative yields (Scheme 26).

Intuitively, lower temperatures (−20 °C) and the use of an apolar solvent (toluene) were mandatory to promote the stereochemical control of the nucleophilic attack when chiral tertiary alcohols **54** were employed as reactants. Here, the enantiospecificity is due to the nucleophilic attack occurring at the opposite face of an oxo-iron intermediate resulting in an inversion of configuration of the stereogenic center (Scheme 27b).

TMs-catalyzed asymmetric hydrogenations are among the most important synthetic transformations and constitute, currently, a routine methodology in industrial organic chemistry [78,79]. In this context, asymmetric transfer hydrogenation (ATH) has played an important role for the synthesis of enantiopure alcohols [80]. Iron-catalyzed ATH has found wide applicability in the development of sustainable hydrogenation reactions mainly due to the low cost with respect to the ruthenium catalysts [81]. Mezzetti and co-workers reported on a (NH)_2_P_2_ macrocyclic-based iron(II) catalysts **L14**·Fe(CNR)_2_(BF_4_)_2_ for the asymmetric hydrogenation of ketones and 1,2-diketones [82,83]. More recently, the authors investigated on the mode of action of complex **L14**·Fe(CNR)_2_(BF_4_)_2_ through NMR spectroscopy and DFT calculations [84]. The catalytic activity of **L14**·Fe(CNR)_2_(BF_4_)_2_ (**T**) and **L14**·FeH(CNR)(BF_4_) (**U**) was tested in the hydrogenation of various ketones in the presence of 2-propanol, with and without the use of additional bases respectively, providing a comparable outcome. In order to elucidate the activation process, complex **T** was titrated with NaOiPr (10 equiv)/*i*PrOH; the resulting equilibrium demonstrated the formation of the hydride complex **U** through intermediate **T’** (Scheme 28a). The reaction of complex **U** with acetophenone at −40 °C enabled the NMR identification of two diastereoisomeric phenylethanolate complexes (a)-**V** and (b)-**V** in the 2.2:1 ratio (Scheme 28b).

The catalytic cycle was investigated by DFT calculations using the hydride complex **U** since it resulted as the most stable species. The lowest energy pathway is depicted below (Scheme 29). Acetophenone approaches complex **U** where the N‒H–O bond directs and activates the carbonyl. The subsequent hydride transfer, which is enantiodetermining, generates the phenylethanoate bonded to intermediate **U’**. The hydride transfer resulted in being barrierless, therefore the enantioselection is determined by the Boltzmann distribution of **TS**-(*b*)-**V** and **TS**-(*a*)-**V**, where the latter is the most stable. This model justifies the high enantioselectivities achieved by the iron-catalyzed ATH. Then, two reaction pathways can be followed: (**a**) a barrierless proton transfer that occurs from the solvent (2-propanol) to the phenylethanoate providing intermediate **U’’’**, or otherwise, (**b**) the formation of an amido-iron complex **U’’**, which engages a new hydrogen bond with 2-propanol (**U’’’**), through product release. A final proton elimination from intermediate **U’’’** forms acetone regenerating catalyst **U** for both depicted pathways. Interestingly, in the catalytic cycle, intermediate **U’’’** resulted the highest in energy and its formation is guaranteed only in the presence of an excess of 2-propanol. Contrarily, the observed (*a*)-**V** and **T’** intermediates were lower in energy with respect to **U’’’**, suggesting these complexes as resting species in the catalytic cycle.

Mechanistic studies on N_2_P_2_ macrocyclic-based iron(II) complexes, active for the asymmetric hydrogenation of ketones, were performed by Zuo [84]. The amido-ene(amido) carbonyl iron(II) complex **L15**·Fe(II)CO was reported to efficiently oxidize 2-propanol through the formation of an amino hydrido iron(II) species **L15**·Fe(II)H(CO) (Scheme 30) [85]. Both iron(II) species are stabilized by covalent π-backdonation interactions between the ene(amido) fragment and the iron center. This stabilization is the reason for the high catalytic activity observed compared to other N_2_P_2_ complexes that do not present the ene(amido) group.

Knölker-type iron complexes are known to efficiently catalyze hydrogen transfer reactions [86]; however, despite their wide application, this class of catalysts suffered from a poor stereocontrol. Recently, the design of a new Knölker-type catalyst **L16**·Fe(0)(CO)_3_, featuring a C2-symmetric cyclopentadienone ligand bearing chiral substituents at the 2 and 5 positions, provided *ees* up to 70% for the asymmetric hydrogenation of different ketones (Scheme 31) [87].

When new chiral ligands are designed for asymmetric transformations, a rigid backbone is usually pursued to ensure a successful chiral induction to the products. Recently, Feng introduced a new class of conformationally flexible, C2 symmetric *N*,*N*′-dioxide chiral ligands, able to promote highly enantiocontrolled catalytic transformations [88]. These ligands display two alkyl amine oxide–amide subunits separated by an alkyl spacer and they are easily synthesized from commercially available, optically pure amino acids. In 2020 Feng’s research group showed the versatility of this class of ligands in combination with iron salts. Indeed, they reported the halohydroxylation of α,β unsaturated ketones **58** using water as a nucleophile [89]. The use of a l-ramipril derivative ligand (**L17**), able to coordinate the Fe(II) center in a tetradentate manner, led to the desired transformation with *ee*s up to 98%. Different substituents on the chalcone derivative were well tolerated; *ortho*-substituted aryls (**59b**) gave lower *dr* than the corresponding *meta*- and *para*-substituted ones, probably due to the increased steric hindrance between the substrates and the catalyst (Scheme 32).

The same research group also reported a chiral *N*,*N*′-dioxide-iron(III) catalyst for the asymmetric oxidation of sulfides to chiral sulfoxides **61** [90]. A high-spin **L18**·Fe(III)-OOH intermediate was proposed to be the most probable active species in the catalytic cycle, while the enantio-discrimination of heterotopic lone pairs, was determined by steric repulsions between sulfides **60** and the spatially encumbered ligands (**L18,19**). The procedure gave access to a broad range of chiral sulfoxides **61**, with high functional group tolerance. The biologically active sulfoxide (*R*)-modafinil (**61a**) was synthesized on a gram scale (Scheme 33).

## 8. Conclusions

The development of more sustainable, cheaper and stereoselective synthetic methodologies are crucial for the future. As emerged by this review, asymmetric iron catalysis represents a clear and potentially ground-breaking alternative to the well-established 4d and 5d metals-catalyzed reactions. This has been made possible by controlling the catalytic activity of iron catalysts through a proper choice of ligands. Particularly, we summarized the recent advances focusing on the mode of action of chiral iron catalysts. Different approaches have been envisioned for controlling the enantioselectivity of the transformation; the most common involve the use of chiral ligands in combination with iron precursors salts. Among them, chiral oxazoline-containing ligands received remarkable attention for the enantioselective construction of carbon-silicon bonds. These ligands proved their efficiency also in cyclopropanation reactions. Another interesting class of emerging ligands is represented by NHCs; in combination with an iron catalyst, they promoted the C‒H hydroarylation of alkenes. Of course, phosphorous containing ligands are still the most utilized for enantiocontrolled reactions. Bidentate chiral phosphines were found active for the enantioconvergent construction of chiral C‒C bonds via Suzuki–Miyaura cross-coupling or for DG-mediated C‒H functionalization reactions. Chiral phosphoric acids, on the other hand, were found suitable for the synthesis of axially chiral molecules derivatives through radical iron catalysis. The same approach could be developed through the aid of chiral diphospine oxide ligands or using chiral auxiliaries. Recently, it has been proved that ligands featuring rigid backbones are not strictly required for ensuring high levels of asymmetric induction. In fact, conformationally flexible C2 symmetric *N*,*N*′-dioxide chiral ligands were found suitable for the asymmetric halohydroxylation of α,β unsaturated ketones as well as for the oxidation of sulfides to chiral sulfoxides. Chiral N_2_P_2_ complexes and Knölker-type catalysts have been extensively investigated for the asymmetric transfer hydrogenation of ketones with a sacrificial hydrogen source. A more recent approach, to promote enantioselective reactions, consists of chiral-at-metal catalysis. In this case, asymmetric coordination of achiral ligands at the iron center generated chiral octahedral iron catalysts. These complexes were found active in the asymmetric Nazarov reaction or FC-type alkylations. Asymmetric transformations could be developed by exploiting second coordination spheres usually consisting of a protein or DNA strands. Different enzymes containing cytochrome P450 were modified to promote nitrene or carbene transfer reactions. Moreover, the DNA/iron-porphyrin catalytic system was found active for the cyclopropanation of alkenes.

In this scenario, many other challenges need to be addressed, particularly with the aim to design new chiral ligands to expand the portfolio of asymmetric iron-catalyzed C‒C bond forming reactions as in example (reductive) cross-coupling [91] or C‒H functionalization reactions. Furthermore, a deeper understanding of reaction mechanism and particularly of the role played by additives and ligands, will be crucial for the application of stereoselective iron-catalyzed transformations in continuous-flow processes [92] or the development of photoredox iron-catalyzed transformations [93].

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
