# Peer review of "Recent Advances in Asymmetric Iron Catalysis"

_molecules, 2020, doi:10.3390/molecules25173889_

Round 1

Reviewer 1 Report

This review describes an interesting analysis in the latest achievements and new concepts brought in the field of enantioselective iron-catalyzed transformations. In general, the manuscript is clear and well organized. In the opinion of this reviewer, some recent contributions are missed, and deserve their inclusion in this review. The manuscript may be considered for publication in Molecules after minor revisions.

Catalytic Asymmetric Halohydroxylation of α,β‐Unsaturated Ketones with Water as the Nucleophile
https://doi.org/10.1002/adsc.202000080
Chiral N,N′-dioxide–iron(iii)-catalyzed asymmetric sulfoxidation with hydrogen peroxide
https://doi.org/10.1039/D0CC00434K

The Stabilization Effect of π-backdonation Ligands on the Catalytic Reactivities of Amido-ene(amido) Iron Catalysts in the Asymmetric Transfer Hydrogenation of Ketones
https://doi.org/10.1002/ejic.202000470

Synthesis and Catalytic Application of Knölker-Type Iron Complexes with a Novel Asymmetric Cyclopentadienone Ligand Design
https://doi.org/10.3390/catal9100790

Additional comments:

Likewise, I can see some grammar and spelling mistakes and some sentences need to be checked. I include some examples but consider reviewing all the manuscript.

Page 2, line 57: In example,… replace by For example,…

Page 4, line 90: (up to 96% ee).. must be replaced by (up to 90% ee)

Page 5, line 104: The prosed… must be replaced by The proposed...

Page 7, line 143:  …transition metals- must be ….transition metal-

Page 11, last paragraph: Revise the grammar and correct the syntax

Page 15, line 301: …and the quinolone ligand moieties reduces…. must be corrected with ….quinoline ligand moieties reduce….

Author Response

please, see the attachement

Reviewer 2 Report

This review summarized the latest advances and provided new concepts in Fe-catalyzed enantioselective transformations, which should be highly interesting to the catalytic and organic community. Furthermore, the manuscript is well organized and written, and the references are fully provided. I therefroe recommend it for the publication in Molecules after the minor revisions as follows:

In references 23, 48 and 56,Journal abreviations should be used.

Author Response

please, see the attachement

Reviewer 3 Report

In this Review, entitled Recent Advances in Asymmetric Iron Catalysis, Cera and Co-workers talk about recent advances regarding the mode of action of iron-based catalysts, highlighting different approaches to controlling the enantioselectivity. The work is well written and clearly lists numerous examples, more or less recent.

Some things to correct:

- in the introduction, line 46-48 the authors write “As a matter of the fact, asymmetric catalysis with transition metals proved to be the most economical, environmentally friendly and efficient approach for the synthesis of enantiomerically pure molecules”. I believe that this sentence should be modified, bearing in mind that organocatalysis exists.

- Scheme 14a, R indicated in compound 22, is to be replaced with iPr; Delta-Fe gives the product (S) and Lambda-Fe gives the product (R), exchange as written now; finally, the products indicated are not 24 but 22

- In the manuscript, I would change L to Lx, when generic ligands are listed, in order not to confuse them with the nomenclature from A to V of catalytic cycles. For example (L)3Fe on page 3, line 208 and in iron-binding active site in Scheme 22

- Scheme 18, 33 is missing on TsN3 (otherwise all numbering changes)

For these reasons I believe that this work is suitable for publication in this journal after minor revisions.

Author Response

please, see the attachment
